# Extracorporeal Shock Wave Therapy (ESWT) vs. Exercise in Thumb Osteoarthritis (SWEX-TO): Prospective Clinical Trial at 6 Months

**DOI:** 10.3390/life14111453

**Published:** 2024-11-08

**Authors:** Ilaria Covelli, Silvana De Giorgi, Antonio Di Lorenzo, Angelo Pavone, Fabrizio Salvato, Francesco Rifino, Biagio Moretti, Giuseppe Solarino, Angela Notarnicola

**Affiliations:** 1Orthopedics Unit, Department of Translational Biomedicine and Neuroscience “DiBraiN”, School of Medicine and Surgery, University of Bari, General Hospital, 70124 Bari, Italy; ilaria.covelli@uniba.it (I.C.); silvana.degiorgi@uniba.it (S.D.G.); a.pavone20@studenti.uniba.it (A.P.); f.salvato1@studenti.uniba.it (F.S.); francesco.rifino@policlinico.ba.it (F.R.); biagio.moretti@uniba.it (B.M.); giuseppe.solarino@uniba.it (G.S.); 2Interdisciplinary Department of Medicine, University of Study of Bari, Piazza Giulio Cesare 11, 70124 Bari, Italy; antonio.dilorenzo@uniba.it

**Keywords:** extracorporeal shock wave therapy, exercise, physical therapy, osteoarthritis, trapeziometacarpal, thumb

## Abstract

Rhizarthrosis is the most frequent form of osteoarthritis and is responsible for pain, disability and reduced functionality. The aim of the study is to investigate the clinical effects of shock wave treatment in patients suffering from arthritis of the first finger. A prospective clinical study was designed, in which 72 patients affected by rhizarthrosis were randomized to treatment with shock waves or exercises; in both groups an immobilization brace was used on the first finger. At recruitment and at 1, 3 and 6 months, patients received assessments of pain (VAS), functionality (FIHOA) and disability (DASH); furthermore, the perception of improvement was monitored during follow-ups (Roles and Maudsley Score). In both groups there was a significant improvement in all scores in the comparison between recruitment and 6 months. The perception of improvement was statistically better in the shock wave group than in the exercise group at 1, 3 and 6 months. The regression analysis showed that the reduction of VAS and DASH were statistically influenced by shock wave treatment (*p* < 0.001). Both therapies are effective in managing patients suffering from arthritis of the first finger, but the shock wave treatment could ensure greater persistence of improvements.

## 1. Introduction

Osteoarthritis (OA) is a degenerative inflammatory joint disease with high prevalence, negative effects on quality of life and high costs to the healthcare system [1].

It has been estimated that over 527 million people suffer from OA worldwide and the incidence of this disease is influenced by many factors, such as work, sports participation, musculoskeletal injuries, obesity and sex [2]. OA of the hands is the most common form and predominantly affects women. It occurs in 21% of the population over the age of 40 and causes pain and disability [3]. Furthermore, it is responsible for deformity, stiffness and reduced mobility and strength, limiting common activities such as opening containers, carrying objects and holding pens.

The treatment of choice for OA of the base of the first toe is conservative, while surgical treatment is reserved for those whose symptoms do not improve [3]. Surgical management, however, may be burdened by a number of complications, such as tendon rupture, sensory alterations and infection [4]. The traditional treatment involves a period of immobilization with a brace with a splint on the first finger, associated with physiotherapy treatment [5]. The use of the brace reduces pain, preserves first web space, stabilizes the base of the first metacarpal during pinching, and prevents adduction of the head of the first metacarpal into the palm of the hand and dorsal subluxation of the base of the metacarpal trapezius [6]. A program of specific exercises for the thenar muscles (traditionally pinching exercises) allows patients to rebalance the deforming force of the trapeziometacarpal joint, in which a strong traction by the adductor muscle of the thumb occurs, combined with the weakness of the intrinsic thenar muscles [5]. Physical therapy may improve pain and function in patients with OA of the base of the thumb [7]. The European League Against Rheumatism (EULAR) guidelines [8] recommend physical treatments, such as ultrasound, laser, analgesic currents and heat therapies. The first preliminary experiences relating to high-energy laser therapy have demonstrated effectiveness in reducing pain at 12 weeks [9]. As regards shock wave therapy, so far, only one work has advanced its potential for the treatment of rhizarthrosis [10].

The aim of this study is to compare the effects of a shock wave treatment compared to a standard treatment with exercises in patients suffering from arthritis of the first finger. Both treatments were associated with the use of a brace.

## 2. Materials and Methods

The present study is a prospective randomized clinical trial to evaluate clinical and functional outcomes and satisfaction in patients affected by osteoarthritis of the first finger of the hand. The Territorial Ethics Committee of the “Consorziale Policlinico” University Hospital authorized the study (Interregional Ethics Committee, approval no. 7814, meeting of 20 December 2023). Participants gave their written informed consent. The trial was registered at https://clinicaltrials.gov/ with the trial registration number NCT06056765.

The patients were examined in the outpatient clinic of the Orthopedics and Traumatology Unit of the Policlinico di Bari, Italy. All eligible patients suffering from pain on the radial side of the carpus, which suggested arthrosis of the trapeziometacarpal joint, were examined by a hand surgeon. If the patient fell within the following stated criteria, they were selected for the study. The inclusion criteria were: trapeziometacarpal arthrosis with stage 1 or 2 of the Eaton–Littler radiographic classification and pain (recent radiograph within 6 months previously) [11]; clinical picture that has been occurring for at least 6 months; pain, counted with the Visual Analogue Scale (VAS), of at least 4/10. The exclusion criteria were: rheumatoid arthritis or results of trauma in the affected area; contra-indications to treatment with Extracorporeal Shock Wave Treatment (ESWT) (neoplasia, pregnancy, thrombocytopenia, epilepsy, uncompensated heart disease or arrhythmia, pacemaker, local infections); corticosteroid infiltration or physical therapy in the previous 4 weeks. Seventy-two consecutive patients were recruited. Patients were randomized to two types of treatment: shock wave therapy (36 patients) (shock wave group) or therapeutic exercise (36 patients) (exercise group). All patients used a one-finger splint.

### 2.1. Shock Wave Group

The therapy was applied using a focused shock wave device (Minilith, Storz, Schaffhausen, Switzerland) at the trapeziometacarpal joint, under ultrasound guidance (Figure 1). The shock wave therapy was performed with the patient’s hand in intermediate prono-supination and was administered once a week for three sessions. For each treatment session, 2000 pulses were applied with an average energy flux density (EDF) between 0.03 and 0.08 mJ/mm^2^ and a frequency of four pulses per second (4 Hz). Gel was used between the probe and the skin during applications to ensure conductivity. No local anesthetic was administered. The energy parameters and the delivery mode were defined in accordance with the literature [12].

### 2.2. Exercise Group Program

Patients were instructed in a stretching, stabilization and strengthening program for the thumbs in the 4 weeks following recruitment [13]. These exercises were aimed at stabilizing the muscles of the thumb, the first dorsal interosseous, the abductor pollicis and flexor pollicis brevis (FPB) muscles. The exercise was initially presented as active movement and, if tolerated, resistance was added. A single physiotherapist was involved for all patients in order to avoid operator bias. The same physiotherapist also provided instructions on the exercise program in the clinic and the subjects performed the exercise sessions at home, 2–3 times a day, every day. Each set of exercises took about 10 min. Once a week, a check-up was performed on the method and frequency of the exercises.

The exercises proposed were the following (Figure 2):Massage the space between the first and second finger (thenar eminence) for 3 min (Figure 2a) [14,15].Extend the space between first and second finger, maintaining for 30 s and perform four times per sitting (Figure 2b) [14].“C” contraction: position the first finger and the remaining four fingers as if one wanted to form a “C”; maintain for 30 s and perform four times per sitting (Figure 2c) [14,15].Active range of motion of the first dorsal interosseous: perform an active radial deviation of the index finger, with the hand resting on the table, for 10 repetitions [14,15,16]. If there was no pain, the next session of the exercise was performed with elastic resistance, asking to gradually increase the size and/or strength of the elastic band as much as possible. If there was pain, the previous movement was repeated (Figure 2d).Active thumb abduction: active thumb abduction to maintain and/or increase the space between the first and second fingers, for 10 repetitions [14,15,17,18]. If the subject was able to complete 10 repetitions without pain, at the next session he was asked to perform this exercise with resistance (manually or through the use of an elastic band). The subject gradually increased the size and/or strength of the rubber band. If there was pain, he returned to the previous movement (Figure 2e).Active flexion of the first finger: performing flexion of the trapeziometacarpal joint, for 10 repetitions [15,19]. If the subject was able to complete 10 repetitions with good technique, resistance was added manually or with rubber bands in the next session. If this exercise was painful, they were asked to return to active movement only (Figure 2f).
Figure 2The six exercises administered to the exercise group. Exercise 1: Massaging the thenar eminence (**a**). Exercise 2: Extension of the space between the first and second finger (**b**). Exercise 3: “C” contraction (**c**). Exercise 4: Active range of motion of the first dorsal interosseous (**d**). Exercise 5: Active thumb abduction (**e**). Exercise 6: Active flexion of the first finger (**f**).
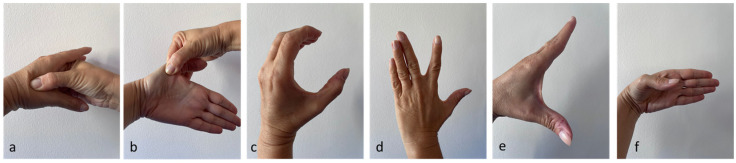


### 2.3. Post-Treatment Care

Patients in both groups were prescribed a full-time brace for 4 weeks to protect the joint and relieve inflammation and pain [20,21]. After this period, they were instructed regarding manual activities that should be avoided in order to prevent recurrence of pain in the trapeziometacarpal joint; these included the exclusion of a strong grip, an imbalance between joint movement and rest, exposure of the finger joints to vibration, and using the joint on an unstable plane [22].

### 2.4. Outcome Measures

The assessments were conducted at the beginning of treatment (T0), at 1 month (T1), at 3 months (T2) and at 6 months (T3). At each time point, VAS, the Functional index for hand osteoarthritis (FIHOA), and Disabilities of the Arm, Shoulder and Hand (DASH) were administered. At T1, T2 and T3 the Roles and Maudsley Score was assessed.

Pain was measured with the VAS scale, which consists of a 10 cm horizontal line (with 0 cm corresponding to no pain and 10 cm corresponding to worst pain ever experienced) [23].Hand functionality was studied with the Functional Index for Hand Osteoarthritis (FIHOA), which is a 10-item questionnaire, with four possible answers for each question (0 = possible without difficulty, 1 = possible slight with difficulty, 2 = possible with important difficulty, 3 = impossible); the score is between 0 (no limitation) and 30 (maximum limitation) [24].The disability is analyzed with the Disabilities of the Arm, Shoulder and Hand (DASH): 11 questions with five possible answers each (1 = No difficulty, 2 = Mild difficulty, 3 = Moderate difficulty, 4 = Severe difficulty, 5 = Unable) and a total score ranging from 11 (no difficulty) to 55 (inability) [25].The Roles and Maudsley Score evaluates the patient’s perception of improvement, from 1 (excellent result with no symptoms following treatment) to 4 (poor, symptoms identical or worse than pre-treatment) [26].

### 2.5. Statistical Analysis

Continuous variables will be presented as mean ± standard deviation and range, while categorical variables will be expressed as proportions. To compare continuous variables between groups, either the independent *t*-test or the Wilcoxon rank-sum test will be utilized, depending on the data distribution. For the comparison of categorical variables between groups, the chi-square test will be employed. For all tests, a two-sided *p*-value < 0.05 was considered an indicator of statistical significance.

### 2.6. Sample Size

To estimate the sample size, we considered a VAS value of 7.5 at enrollment (T0) for both groups, with an average reduction at the primary endpoint (6 months) of 4.8 in the control group [13] and 3.8 in the treatment group [10], with a standard deviation of 1.4 for both groups. Sample size estimation was conducted using a *t*-test, with a significance level (alpha) set at 0.05 and a test power of 80%. The estimated sample size was 64 subjects, and to account for a potential 15% loss at follow-up, we aimed to recruit 72 subjects (36 per group). This effect was selected as the smallest effect that would be important to detect, meaning that a smaller effect would not have clinical or substantive significance.

The database was built via Microsoft Excel^®^ 2019. The assignment to the groups was performed through randomization, ensuring homogeneity between the two groups for covariates such as sex and age. Due to the apparent nature of the two treatments, blinding was not possible at a patient and physician level; assessors, however, were blind to the patients’ randomization status.

Randomization was conducted using Stata MP17^®^ software; the randomized patient list was generated in a pseudo-anonymous form, with a code linking each patient to their randomization ID in order to link each subject to their specific group and to their own record. All calculations were performed via Stata MP17^®^ software.

## 3. Results

### 3.1. Population Characteristics

The study enrolled 72 subjects, evenly distributed between the study group and control group (Figure 3).

The population’s characteristics are stated in Table 1 and Table 2. The mean age at diagnosis was 65.40 years (±8.49 years), while the mean Body Mass Index (BMI) was 25.02 kg/m^2^ (±4.14 kg/m^2^). The mean time since the beginning of arthritic symptoms was 9.18 months (±2.41 months).

### 3.2. Endpoints

The VAS, FIHOA, DASH and Roles and Maudsley score values over time are described in Table 3. Normality was proven for the distribution of the VAS values over time. All other scores showed non-normal distribution in the study population. Therefore, the latter were studied via the Wilcoxon signed-rank test.

The mean VAS score showed a significant decrease from T0 to T3 in both the shock wave group (t: 12.80; *p*-value < 0.001) and the exercise group (t: 15.29; *p*-value < 0.001), with a mean reduction of 4.81 (±2.25) in the shock wave group and 2.75 (±1.08) in the exercise group. When considering the intermediate checkpoints, both groups showed a significant reduction of the VAS score from T0 to T1 and from T1 to T2. For the shock wave group, in particular, the T0–T1 interval highlighted a mean 3.39 (±1.76) decrease (t: 11.55; *p*-value < 0.001) and the T1–T2 interval had a mean decrease of 0.72 (±1.14) (t: 3.81; *p*-value < 0.001). For the exercise group, the T0–T1 interval’s VAS score reduction was 1.83 (±0.74) (t: 14.93; *p*-value < 0.001), while the T1–T2 interval had a mean 1.28 (±1.00) decrease (t: 7.64; *p*-value < 0.001). However, when the last interval was considered, a significant 0.69 (±1.01) decrease of the VAS score was highlighted in the shock wave group only (t: 4.13; *p*-value < 0.001), while the exercise group showed a slight, yet significant 0.36 (±1.25) increase of this score (t: −1.74; *p*-value: 0.045).

The FIHOA score showed a significant reduction over time in both groups. The decrease was 7.08 (±6.03) in the shock wave group (z: 5.12; *p*-value < 0.001) and 5.28 (±3.82) in the exercise group (z: 5.14; *p*-value < 0.001). Analyzing the single checkpoints, a consistently significant reduction was identified in the shock wave group for both the T0-T1 (mean: 5.75 ± 5.23; z: 5.03; *p*-value < 0.001) and T1–T2 interval (mean: 0.83 ± 1.71; z: 3.23; *p*-value: 0.001) and in the exercise group for the same intervals (T0-T1 mean: 4.64 ± 3.36; z: 5.22; *p*-value < 0.001; T1-T2 mean: 1.19 ±1.51; z: 4.32; *p*-value < 0.001). However, both groups showed no significant difference between the T2 and T3 values of the FIHOA score; the shock wave group, in particular, had a non-significant 0.5 (±1.95) decrease (z: 1.90; *p*-value: 0.06), while the exercise group had a non-significant 0.56 (±2.21) increase (z: −1.42; *p*-value: 0.15).

A significant reduction of the DASH score was observed from T0 to T3. In the shock wave group, it was a 13.78 (±7.52) decrease (z: 5.24; *p*-value < 0.001), while in the exercise group it was a 10.03 (±4.35) decrease (z: 5.23; *p*-value < 0.001). When breaking the analysis down to the intermediate checkpoints, the study group showed a significant 11.69 (±7.11) reduction of the DASH score from T0 to T1 (z: 5.23; *p*-value), the T1–T2 interval showed a significant 0.89 (±3.97) decrease (z: 3.01; *p*-value < 0.01), and the T2–T3 interval highlighted a significant 1.19 (±2.42) reduction (z: 3.40; *p*-value < 0.001). The exercise group, on the other hand, showed a significant 6.56 (±3.42) decrease of the DASH score (z: 5.23; *p*-value < 0.001) from T0 to T1, followed by a still significant 4.58 (±2.68) T1–T2 decrease (z: 5.15; *p*-value < 0.001); after the T2-T3 interval, however, a significant 1.11 (±3.03) increase of the DASH score was observed (z: −2.07; *p*-value: 0.04).

### 3.3. Inferential Statistics

The VAS modification over time was significantly different between the shock wave and exercise groups (t: −4.94; *p*-value < 0.001). The DASH score reduction was also significantly different between the two groups (z: −2.70; *p*-value < 0.01). On the other hand, the FIHOA score changes did not significantly differ between the shock wave and exercise group (z: −0.98; *p*-value: 0.324). As far as the Roles and Maudsley score is concerned, the shock wave group showed a consistently lower score than the exercise one. The difference was significant at both T1 (z: 3.69; *p*-value < 0.001), T2 (z: 3.32; *p*-value < 0.001) and T3 (z: 4.48; *p*-value < 0.001).

### 3.4. Regression Analysis

The VAS variation over time showed to be significantly impacted by the use of shock wave therapy, with a greater decrease in the shock wave group than in the exercise one (aOR: 2.14; 95% CI: 1.20–3.08; *p*-value < 0.001). The starting value of VAS was also directly associated with the score’s reduction over time (aOR: 0.44; 95% CI: 0.07–0.81; *p*-value: 0.019). Both associations were confirmed by univariable regression (aOR for shock wave therapy: 2.06; 95% CI: 1.22–2.89; *p*-value < 0.001; aOR for VAS at T0: 0.66; 95% CI: 0.31–1.01; *p*-value < 0.001).

The reduction of the FIHOA score was not significantly impacted by the use of shock wave therapy (aOR: 1.93; 95% CI: −0.06–3.92; *p*-value: 0.057). However, a significant direct association was shown for this score’s value at T0 (aOR: 0.59; 95% CI: 0.40–0.78; *p*-value < 0.001), and a reverse association was highlighted for the Ray-X stage (aOR: −2.18; 95% CI: −4.29–−0.07; *p*-value: 0.044). The impact of the T0 FIHOA score was further proved by the univariable regression (aOR: 0.63; 95% CI: 0.46–0.80; *p*-value < 0.001); the Ray-X stage, on the contrary, was not confirmed to be significant impactful on FIHOA score changes over time (aOR: −2.10; 95% CI: −4.75–0.54; *p*-value: 0.117).

The use of shock wave therapy showed a significant association with the DASH score decrease over time (aOR: 3.47; 95% CI: 0.98–5.96; *p*-value < 0.01). The DASH value at T0 was also significantly associated with the score’s decrease (aOR: 0.74; 95% CI: 0.52–0.96; *p*-value < 0.001). Both results were confirmed by the univariable regression (aOR for shock wave therapy: 3.75; 95% CI: 0.86–6.64; *p*-value: 0.012; aOR for the DASH score at T0: 0.73; 95% CI: 0.53–0.93; *p*-value < 0.001).

Finally, the Roles and Maudsley score showed to be mainly influenced by the use of shock wave therapy. The shock wave group, in fact, showed a consistently lower Roles and Maudsley score at all checkpoints (aOR at T1: −0.69; 95% CI: −1.04–−0.34; *p*-value < 0.001; aOR at T2: −0.66; 95% CI: −1.02–−0.31; *p*-value < 0.001; aOR at T3: −1.01; 95% CI: −1.39–−0.62; *p*-value < 0.001). Moreover, the score was significantly higher in subjects whose symptoms had started since a longer period of time both at T2 (aOR: 0.83; 95% CI: 0.01–0.16; *p*-value: 0.029) and T3 (aOR: 0.08; 95% CI: 0.01–0.16; *p*-value: 0.041). However, univariable regression did not confirm the impact of time since symptom onset on the Roles and Maudsley score (aOR at T2: 0.03; 95% CI: −0.04–0.10; *p*-value: 0.376; aOR at T3: 0.02; 95% CI: −0.07–0.10; *p*-value: 0.681). On the contrary, the reverse association of shock wave therapy with Roles and Maudsley score at T1 (aOR: −0.64; 95% CI: −0.94–−0.34; *p*-value < 0.001), T2 (aOR: −0.58; 95% CI: −0.90–−0.26; *p*-value: 0.001) and T3 (aOR: −0.89; 95% CI: −1.24–−0.53; *p*-value < 0.001) was confirmed by univariable regression. The regression analyses results are summarized in Table 4.

## 4. Discussion

This study prospectively compared the benefits of conservative treatment with shock waves and bracing versus exercise and bracing, in patients with early stages of arthritis of the trapeziometacarpal joint (Eaton stages 1–2) [11]. In both groups there was a significant improvement in the assessed parameters of pain (VAS), function (FIHOA) and disability (DASH), in the comparison between recruitment and 6-month follow-up. In the two groups these improvements were also confirmed in the comparison between recruitment and 1 month and between 1 month and 3 months for all scores administered; regarding the comparison between 3 and 6 months, the improvement was maintained only in the shock wave group for pain (VAS) and disability (DASH). In both groups the FIHOA did not present statistically significant improvements between 3 and 6 months. The Roles and Maudsley score presented statistically lower values in the shock wave group than in the exercise group at the three examination times.

In recent years, shock wave therapy has found application in many musculoskeletal pathologies [27]. Shock wave therapy is a physical therapy that exploits the biological effects of an acoustic wave that is focused in a small treatment area and causes cavitation effects [28]. This therapy has proven to be very effective in reducing pain and aiding the functional recovery of tendon pathologies, such as calcific cuff tendinopathy, tennis elbow syndrome, plantar fasciitis and tendon disease; it also allows the healing of fracture nonunions, bone edema and complex regional pain syndrome [28]. A recent meta-analysis demonstrated that the application of ESWT in patients suffering from knee osteoarthritis leads to statistically significant improvement in pain and functional recovery compared to other conservative therapy options [12]. The physical stimulus of shock waves determines an upregulation of various growth factors, modulation of inflammatory cytokines, chemotaxis of stem cells, proliferative effect and revascularizing action [29,30]. When shock waves are applied to the subchondral bone and articular cartilage, neovascularization, osteogenesis and chondrogenesis would occur [30,31,32,33,34,35,36]. On arthritic joints in an animal model, shock waves were able to determine motor recovery [37,38]. To date, only one clinical experience of the application of shock waves in the treatment of rhizarthrosis has been published [10]. Ioppolo and colleagues [10] conducted a clinical study in which they compared the effects of shock waves (SW) vs hyaluronic acid infiltration in the treatment of rhizarthrosis and the results were evaluated at the end of treatment and at 3 and 6 months. Pain reduction and functional recovery were found in both groups. In particular, as regards primary end points, pain, measured with the VAS scale, was significantly reduced by follow up in the two groups, with results in favor of shock waves at the end of treatment and at 6 months. As regards the Duruoz Hand Index functional scale, a functional recovery was recorded in both groups at the different FUs, without significant differences between the two treatments. As regards the secondary end points, in the shock wave group significant improvements were recorded in finger pinch strength at the end of the treatment and in hand grip strength at 6 months, without significant differences between the two groups The authors conclude that in the treatment of rhizarthrosis, ESWT could have superior effects to that of hyaluronic acid infiltration with regards to pain reduction, while it would appear to have similar effects with regards to functional and strength recovery. These results would be consistent with the analgesic and anti-inflammatory effects of ESWT and viscosupplementation of intra-articular HA injections.

In our experience the effect of shock waves has been compared with therapeutic exercise. Both therapies behave like mechanical stimuli, with biological effects on tissues. In particular, joint mobilization would determine beneficial effects both through biomechanical responses and neurophysiological effects [39,40]. In fact, movement determines the release of endorphins and substance P with inhibition of the nociceptive pathways. Therefore, the combined treatment of exercise and bracing provides good results in terms of clinical-functional recovery and stabilization of rhizarthrosis [14,16,41,42,43]. Studies demonstrate that the trapeziometacarpal joint receives its stability from the opposing thumb muscle, abductor pollicis, and the first interosseous [14,16,41,42,43].

Previous randomized clinical trials have shown that effective treatment must be continued for at least 2-4 weeks [43,44,45]. Pisano and colleagues [13] randomized 190 patients suffering from rhizarthrosis to a standard treatment or associated with home exercises and found clinical and functional improvement at 12 months, without statistically significant differences between the two groups. The results made it possible to confirm that the exercises allow the metacarpal trapezius joint to be stabilized, reducing pain and disability. On the other hand, increasing frequency with the addition of home exercises did not bring additional improvement.

The strength of the study is that it is the first study to investigate shock wave treatment in patients suffering from rhizarthrosis in comparison with therapeutic exercise. This non-invasive therapeutic option has proven to be equivalent and could be a better alternative to exercise in patients with initial arthritis of the first finger, in particular with greater persistence of results at 6 months. Longer-term monitoring may allow us to verify the persistence of the benefits, as well as the opportunity to repeat a second cycle of shock waves and/or integration with other conservative treatments.

There are some limitations of the study that need to be considered. We did not use instrumental follow-up controls (X-ray, ultrasound, MRI), which could have provided an indication of the effects of the treatment on the cartilage, bone and muscle-tendon of the trapeziometacarpal joint. Furthermore, the brace may have been under-used or over-used by the patient. Furthermore, in relation to the type of treatment, the blinding of patients and professionals is lacking. There is no control group to distinguish whether the orthosis determines different effects compared to other therapies, considering that the prolonged use of an orthosis improves pain. We cannot eliminate the possibility that this could influence the results. We did not measure grip strength, hypothesizing that the variations in this parameter could be more relevant to the control group, which performed exercises. Another limitation of this study is that, despite periodic checks, we are not sure about the therapeutic adherence of the patients in the exercise group. Furthermore, there is no placebo group, and the number may be relatively low.

Finally, as a perspective for future sub-analysis, pain could be studied via categorization rather than numerical scales. In fact, categories may improve understanding of shockwave therapy on pain from a practical point of view. However, we chose to treat pain as a quantitative variable rather than a qualitative one for statistical purposes, in order to make it behave similarly to other endpoints and make the analysis more homogeneous.

## 5. Conclusions

In conclusion, this study offers a new treatment approach for rhizarthrosis in the early stage. Subsequent studies will be able to evaluate the effectiveness of a combined shock wave treatment, brace, and therapeutic exercise. The combined treatment could offer additional clinical benefits, improving tissue trophism and functional recovery, with longer duration of benefits. Longer-term monitoring will allow us to identify the persistence times of the benefits found.

## Figures and Tables

**Figure 1 life-14-01453-f001:**
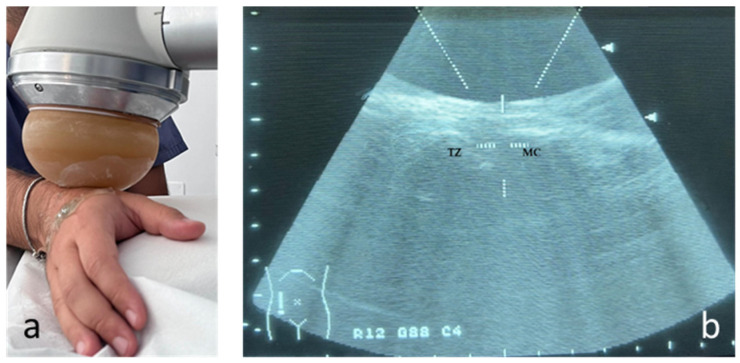
Treatment of osteoarthritis of the trapeziometacarpal joint during shock wave therapy under ultrasound guidance. The positioning of the probe on the wrist during treatment (**a**) and ultrasound image of the district according to the long axis (TZ: trapezius; MC: first metacarpal), with the focal area of treatment identified (white cross) (**b**).

**Figure 3 life-14-01453-f003:**
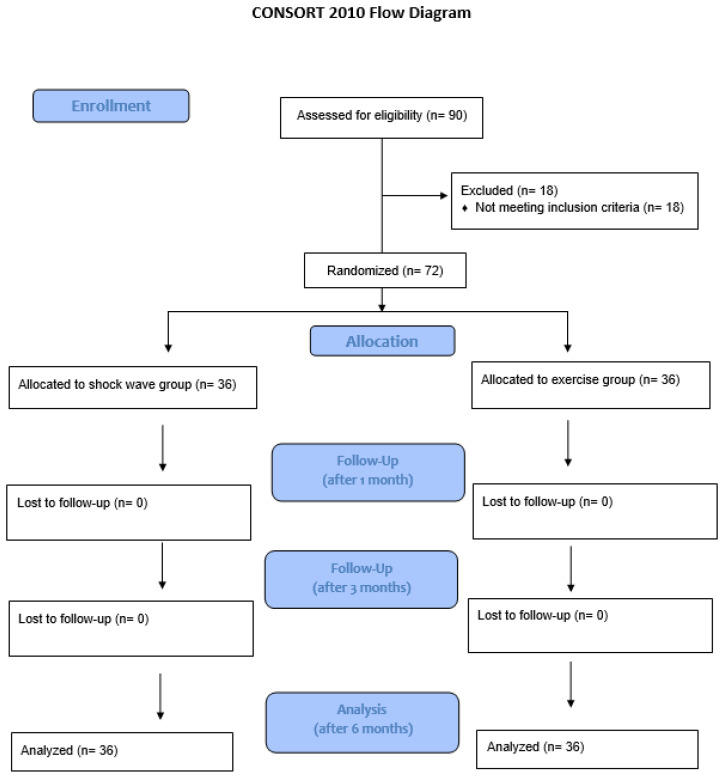
CONSORT flow diagram of participants throughout the study period.

**Table 1 life-14-01453-t001:** Study population characteristics: quantitative variables (mean ± standard deviation).

	Shock Wave Group	Exercise Group	Overall
Age (years)	65.08 ± 8.39	65.72 ± 8.69	65.40 ± 8.49
BMI (kg/m^2^)	25.59 ± 3.28	24.45 ± 3.28	25.05 ± 4.14
Time since onset (months)	10 ± 2.35	8.36 ± 2.22	9.18 ± 2.42

**Table 2 life-14-01453-t002:** Study population characteristics: qualitative variables.

		Shock Wave Group	Exercise Group	Overall
Sex	Female	18	18	36
	Male	18	18	36
Age	<65	18	18	36
	≧65	18	18	36
Dominant hand	Left	2	3	5
	Right	34	33	67
Arthrosis laterality	Left	13	6	19
	Right	23	30	53
Smoking habit	No	34	33	67
	Yes	2	3	5
Hypertensive cardiopathy	No	26	27	53
	Yes	10	9	19
Metabolic/endocrine diseases	No	28	33	61
	Yes	8	3	9
Previous physiotherapy	No	26	29	55
	Yes	10	7	17
Radiological stage of disease	Stage 1	11	9	20
	Stage 2	25	27	52

**Table 3 life-14-01453-t003:** Description of the main endpoints over time; the VAS score is described as median (interquartile range), while other endpoints are described as mean ± standard deviation.

	T0	T1	T2	T3
	**VAS**			
Shock wave group	7 (6–8)	4 (2–4.5)	3.5 (2–4)	2 (0–3)
Exercise group	6 (6–8)	4.5 (4–5)	2.5 (2–5)	4 (3–5)
Overall	7 (6–8)	4 (3–5)	3 (2–4)	3 (2–4.5)
	**FIHOA**			
Shock wave group	13.64 ± 6.39	7.89 ± 5.10	7.06 ± 5.18	6.56 ± 5.28
Exercise group	12.64 ± 4.04	8.00 ± 2.97	6.81 ± 2.71	7.36 ± 3.10
Overall	13.14 ± 5.33	7.94 ± 4.14	6.93 ± 4.11	6.96 ± 4.32
	**DASH**			
Shock wave group	28.28 ± 6.70	16.58 ± 5.07	15.69 ± 6.64	14.50 ± 5.86
Exercise group	27.33 ± 4.68	20.78 ± 5.01	16.19 ± 4.43	17.31 ± 3.61
Overall	27.81 ± 5.76	18.68 ± 5.43	15.94 ± 5.61	15.90 ± 5.03
	**Roles and Maudsley**			
Shock wave group		1.94 ± 0.71	1.75 ± 0.73	1.53 ± 0.77
Exercise group		2.58 ± 0.55	2.33 ± 0.63	2.42 ± 0.73
Overall		2.26 ± 0.71	2.04 ± 0.74	1.97 ± 0.87

**Table 4 life-14-01453-t004:** Summary of multivariable and univariable regression analyses results. Statistically significant associations are marked with an asterisk (*).

**ΔVAS**				
**Multivariable Regression**				
**Independent Variable**	**aOR**	**95% CI Lower Limit**	**95% CI Upper Limit**	***p*-Value**
Shock wave therapy *	2.14	1.20	3.08	<0.001
VAS at T0 *	0.44	0.07	0.81	0.019
Sex (aOR for males vs. females)	0.37	−0.51	1.25	0.401
Age at diagnosis	0.02	−0.26	0.07	0.333
Right-hand dominance	−1.43	−3.25	0.38	0.119
BMI	−0.01	−0.12	0.12	0.968
Smoking habit	−0.14	−1.77	1.49	0.862
Affected hand (aOR for right hand vs. left hand)	0.44	−0.57	1.45	0.385
Months since symptoms’ onset	−0.15	−0.35	0.04	0.114
Hypertensive cardiopathy	−0.22	−1.23	0.79	0.664
Endocrine diseases	0.43	−0.88	1.74	0.512
FANS therapy	0.21	−0.87	1.30	0.698
Previous physiotherapy	−0.14	−1.17	0.89	0.788
X-ray stage	−0.29	−1.30	0.73	0.571
**Univariable Regression**				
**Independent Variable**	**aOR**	**95% CI Lower Limit**	**95% CI Upper Limit**	***p*-Value**
Shock wave therapy *	2.06	1.22	2.89	<0.001
VAS at T0 *	0.66	0.31	1.01	<0.001
**ΔFIHOA**				
**Multivariable Regression**				
**Independent Variable**	**aOR**	**95% CI Lower Limit**	**95% CI Upper Limit**	***p*-Value**
Shock wave therapy	1.93	−0.06	3.92	0.057
FIHOA at T0 *	0.59	0.40	0.78	<0.001
Sex (aOR for males vs. females)	1.08	−0.85	3.01	0.266
Age at diagnosis	−0.3	−0.14	0.07	0.518
Right-hand dominance	−2.18	−6.07	1.71	0.265
BMI	−0.12	−0.36	0.13	0.333
Smoking habit	−2.73	−6.18	0.72	0.118
Affected hand (aOR for right hand vs. left hand)	1.34	−0.84	3.52	0.223
Months since symptoms’ onset	−0.31	−0.72	0.10	0.133
Hypertensive cardiopathy	−0.77	−2.92	1.37	0.473
Endocrine diseases	0.65	−2.20	3.50	0.649
FANS therapy	0.35	−1.95	2.65	0.760
Previous physiotherapy	−0.54	−2.78	1.70	0.632
X-ray stage *	−2.18	−4.29	−0.07	0.044
**Univariable Regression**				
**Independent Variable**	**aOR**	**95% CI Lower Limit**	**95% CI Upper Limit**	***p*-Value**
Shock wave therapy *	0.63	0.46	0.80	<0.001
X−ray stage	−2.10	−4.75	0.54	0.117
**ΔDASH**				
**Multivariable Regression**				
**Independent Variable**	**aOR**	**95% CI Lower Limit**	**95% CI Upper Limit**	***p*-Value**
Shock wave therapy *	3.47	0.98	5.96	0.007
DASH at T0 *	0.74	0.52	0.96	<0.001
Sex (aOR for males vs. females)	1.23	−1.28	3.74	0.331
Age at diagnosis	0.03	−0.10	0.17	0.633
Right-hand dominance	−1.14	−6.11	3.82	0.647
BMI	0.14	−0.17	0.46	0.360
Smoking habit	−2.07	−6.47	2.33	0.350
Affected hand (aOR for right hand vs. left hand)	1.09	−1.67	3.86	0.431
Months since symptoms’ onset	−0.34	−0.87	0.18	0.200
Hypertensive cardiopathy	−2.07	−4.80	0.66	0.135
Endocrine diseases	1.71	−1.86	5.29	0.341
FANS therapy	0.26	−2.73	3.24	0.863
Previous physiotherapy	−0.94	−3.77	1.88	0.506
X-ray stage *	−1.35	−4.17	1.46	0.341
**Univariable Regression**				
**Independent Variable**	**aOR**	**95% CI Lower Limit**	**95% CI Upper Limit**	***p*-Value**
Shock wave therapy *	3.75	0.86	6.64	0.012
DASH at T0 *	0.73	0.53	0.93	<0.001
**R&M at T1**				
**Multivariable Regression**				
**Independent Variable**	**aOR**	**95% CI Lower Limit**	**95% CI Upper Limit**	***p*-Value**
Shock wave therapy *	−0.69	−1.03	−0.34	<0.001
Sex (aOR for males vs. females)	0.10	−0.23	0.44	0.537
Age at diagnosis	0.01	−0.01	0.02	0.527
Right-hand dominance	0.20	−0.50	0.90	0.574
BMI	−0.02	−0.92	0.32	0.279
Smoking habit	−0.30	−0.92	0.32	0.332
Affected hand (aOR for right hand vs. left hand)	−0.08	−0.47	0.30	0.671
Months since symptoms’ onset	0.05	−0.02	0.13	0.131
Hypertensive cardiopathy	0.13	−0.25	0.52	0.491
Endocrine diseases	−0.43	−0.94	0.07	0.090
FANS therapy	−0.05	−0.46	0.36	0.813
Previous physiotherapy	0.15	−0.25	0.55	0.462
X-ray stage	−0.12	−0.49	0.26	0.530
**Univariable Regression**				
**Independent Variable**	**aOR**	**95% CI Lower Limit**	**95% CI Upper Limit**	***p*-Value**
Shock wave therapy *	−0.64	−0.94	−0.34	<0.001
**R&M at T2**				
**Multivariable Regression**				
**Independent Variable**	**aOR**	**95% CI Lower Limit**	**95% CI Upper Limit**	***p*-Value**
Shock wave therapy *	−0.66	−1.02	−0.31	<0.001
Sex (aOR for males vs. females)	−0.15	−0.50	0.19	0.376
Age at diagnosis	0.01	−0.01	0.03	0.318
Right-hand dominance	0.07	−0.65	0.78	0.854
BMI	−0.02	−0.06	0.03	0.415
Smoking habit	−0.15	−0.78	0.48	0.634
Affected hand (aOR for right hand vs. left hand)	−0.01	−0.40	0.39	0.995
Months since symptoms’ onset *	0.08	0.01	0.16	0.029
Hypertensive cardiopathy	−0.10	−0.49	0.30	0.620
Endocrine diseases	−0.31	−0.83	0.20	0.226
FANS therapy	−0.31	−0.74	0.11	0.142
Previous physiotherapy	0.26	−0.15	0.67	0.205
X-ray stage	−0.01	−0.39	0.37	0.956
**Univariable Regression**				
**Independent Variable**	**aOR**	**95% CI Lower Limit**	**95% CI Upper Limit**	***p*-Value**
Shock wave therapy *	−0.58	−0.90	−0.26	0.001
Months since symptoms’ onset	0.32	−0.04	0.10	0.376
**R&M at T3**				
**Multivariable Regression**				
**Independent Variable**	**aOR**	**95% CI Lower Limit**	**95% CI Upper Limit**	***p*-Value**
Shock wave therapy *	−1.01	−1.39	−0.62	<0.001
Sex (aOR for males vs. females)	−0.06	−0.44	0.32	0.758
Age at diagnosis	−0.01	−0.02	0.02	0.877
Right-hand dominance	0.25	−0.54	1.03	0.531
BMI	−0.04	−0.09	0.01	0.078
Smoking habit	−0.13	−0.82	0.56	0.716
Affected hand (aOR for right hand vs. left hand)	−0.40	−0.83	0.03	0.067
Months since symptoms’ onset *	0.08	0.01	0.16	0.041
Hypertensive cardiopathy	0.03	−0.40	0.46	0.898
Endocrine diseases	−0.45	−1.02	0.11	0.110
FANS therapy	−0.17	−0.63	0.29	0.458
Previous physiotherapy	0.12	−0.38	0.46	0.582
X-ray stage	0.04	−0.38	0.46	0.843
**Univariable Regression**				
**Independent Variable**	**aOR**	**95% CI Lower Limit**	**95% CI Upper Limit**	***p*-Value**
Shock wave therapy *	−0.89	−1.24	−0.53	<0.001
Months since symptoms’ onset	0.02	−0.07	0.10	0.681

## Data Availability

The datasets generated and/or analyzed during the current study are available from the corresponding author upon reasonable request due to privacy.

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
