# Peer review of "Extracorporeal Shock Wave Therapy (ESWT) vs. Exercise in Thumb Osteoarthritis (SWEX-TO): Prospective Clinical Trial at 6 Months"

_life, 2024, doi:10.3390/life14111453_

Round 1

Reviewer 1 Report

Comments and Suggestions for Authors

Your study provides valuable data for clinicians dealing with thumb OA. The comparison of this two therapies, shock wave therapy and exercise offers practical insights. Despite a few limitations, the findings are significant to this field and add important knowledge for non-surgical treatment methods for thumb base osteoarthritis.

Author Response

Reviewer n. 1

Comments and Suggestions for Authors

Your study provides valuable data for clinicians dealing with thumb OA. The comparison of this two therapies, shock wave therapy and exercise offers practical insights. Despite a few limitations, the findings are significant to this field and add important knowledge for non-surgical treatment methods for thumb base osteoarthritis.

Answer: We appreciate all the comments that allow us to improve the text of the article. Changed parts in the text are highlighted in green.

Reviewer 2 Report

Comments and Suggestions for Authors

General evaluation and characteristics of the reviewed article:

The article Extracorporeal Shock Waves Therapy (ESWT) vs Exercise in Thumb Osteoarthritis (SWEX-TO): prospective clinical trial at 6 months presents a prospective clinical trial comparing the efficacy of Extracorporeal Shock Wave Therapy (ESWT) and exercise in treating rhizarthrosis (thumb osteoarthritis). Seventy-two patients were randomized into ESWT and exercise groups, with both groups using an immobilization brace. Assessments of pain (VAS), functionality (FIHOA), disability (DASH), and perceived improvement (Roles and Maudsley Score) were conducted at baseline, 1, 3, and 6 months.

Results indicated significant improvements in all scores for both groups over the 6-month period. However, the ESWT group demonstrated superior outcomes in pain relief, functionality, and disability reduction, particularly at the 6-month mark. The perception of improvement was statistically better in the ESWT group at all time points. Regression analysis showed that ESWT significantly influenced the reduction of pain and disability scores (p<0.001).

The study concludes that while both ESWT and exercise with bracing are effective, ESWT provides additional benefits in managing thumb osteoarthritis. The article is well-structured with clear presentation of methods and results, offering valuable evidence for ESWT as a promising non-invasive treatment option. Future research should explore the long-term effects of ESWT beyond 6 months and its efficacy in other forms of osteoarthritis.

The paper is interesting, addressing a highly significant and current topic, and is generally well-written. However, before further processing and acceptance, it requires minor corrections and additions. Below are my detailed comments and suggestions.

Minor comments:

The article was not adapted to the journal's requirements and formatting, probably by accident the authors posted the version submitted earlier.

The abstract is far too long, please rebuild it by keeping only the most important information. The abstract should be adapted to the requirements of the journal.

Please expand on the osteoarthritis information in the introduction. This will allow you to better introduce the topic and emphasize the importance of the problem. The incidence of osteoarthritis is influenced by many factors, such as work, sports participation, musculoskeletal injuries, obesity and gender. Information about this, along with the necessary literature, should be added to the first paragraph of the introduction. I suggest adding the following references to this paragraph:

https://doi.org/10.3390/healthcare12161648

DOI: 10.1056/NEJMcp1903768

Figure 1 is not very legible, and I recommend changing its quality and including a corrected version in the text.

The presentation of statistical analysis results is unclear. Please also add tabular summaries with analysis results. This will make it easier to interpret the overall results. 

After implementing the revisions and completing both the text and references, the work will certainly be ready for acceptance in the journal Life.

Author Response

Reviewer n. 2

Minor comments:

Comment 1:

The article was not adapted to the journal's requirements and formatting, probably by accident the authors posted the version submitted earlier. The abstract is far too long, please rebuild it by keeping only the most important information. The abstract should be adapted to the requirements of the journal.

answer 1: Yes, I did.

Comment 2:

Please expand on the osteoarthritis information in the introduction. This will allow you to better introduce the topic and emphasize the importance of the problem. The incidence of osteoarthritis is influenced by many factors, such as work, sports participation, musculoskeletal injuries, obesity and gender. Information about this, along with the necessary literature, should be added to the first paragraph of the introduction. I suggest adding the following references to this paragraph:

https://doi.org/10.3390/healthcare12161648

DOI: 10.1056/NEJMcp1903768

answer 2: Yes, I did.

Comment 3: Figure 1 is not very legible, and I recommend changing its quality and including a corrected version in the text.

Answer 3: Yes, I did.

Comment 4: The presentation of statistical analysis results is unclear. Please also add tabular summaries with analysis results. This will make it easier to interpret the overall results. 

Answer 4: Thank you for the input. We added a table summarizing the results of regression analysis.

After implementing the revisions and completing both the text and references, the work will certainly be ready for acceptance in the journal Life.

Reviewer 3 Report

Comments and Suggestions for Authors

Good study to address the ESWT vs Exercise in Thumb Osteoarthritis.

 1. In SHOCK WAVE GROUP, patients received once a week, for 3 sessions, means three treatment in 3 weeks in total? Any specific reason to apply 3 times in total? How about increase to 6 times, would it be more efficient?

2. For EXERCISE GROUP PROGRAM, I believe 4 weeks exercise is not match the 3 weeks programs in SHOCK WAVE GROUP.

3. Statistical analysis please address the setting P value.

4. Could authors also present a figure to show the results in 3.4. Regression analysis?

5. Please add comment about the combined effect of two treatment on  Thumb Osteoarthritis.

Author Response

Reviewer n. 3

Good study to address the ESWT vs Exercise in Thumb Osteoarthritis.

Comment 1. In SHOCK WAVE GROUP, patients received once a week, for 3 sessions, means three treatment in 3 weeks in total? Any specific reason to apply 3 times in total? How about increase to 6 times, would it be more efficient?

Answer 1: Treatments with focused shock waves have now been standardized in three sessions. In the past, a greater number of sessions were administered, but the effects are comparable to treatments of three sessions and the treatments were standardized to three sessions. In treatments with radial waves the number of sessions is greater, on average between 6 and 10 sessions. In this study we administered focused shock wave in three total sessions, with a frequency of one session per week.

Comment 2. For EXERCISE GROUP PROGRAM, I believe 4 weeks exercise is not match the 3 weeks programs in SHOCK WAVE GROUP.

Answer 2: In our study the population was randomized to perform one of the two treatments, exercises or shock waves. The treatment with exercises was scheduled for 4 weeks, in accordance with the protocols described in the literature. The treatment with focused shock waves was carried out by planning one session per week for three weeks. The patients were re-evaluated at 1, 3 and 6 months from recruitment. The study therefore included an experimental group (the treatment with shock waves) compared with a control group (therapeutic exercises). The two methods, although differing in type of stimulation, biostimulation in the case of shock waves and rehabilitation in the exercise group, can allow us to verify whether a new treatment can be adequately effective as a treatment already described by the guidelines. At the first follow-up, patients in both groups had completed the two therapies and at subsequent follow-ups it was possible to verify whether the improvements were maintained, increased or lost.

Comment 3. Statistical analysis please address the setting P value.

Answer 3: Thank you for noticing, we added the p-value for stastical significance.

Comment 4. Could authors also present a figure to show the results in 3.4. Regression analysis?

Answer 4 Thank you for the input. We added a table summarizing the results of regression analysis, since various reviewers suggested to.

Comment 5. Please add comment about the combined effect of two treatment on Thumb Osteoarthritis.

Answer 5: Yes, I did. I introduced these senctences “Subsequent studies will be able to evaluate the effectiveness of a combined shock wave treatment, brace and therapeutic exercise. The combined treatment could offer additional clinical benefits, improving tissue trophism and functional recovery, with longer duration of benefits. Longer-term monitoring will allow us to identify the persistence times of the benefits found”

Reviewer 4 Report

Comments and Suggestions for Authors

1. Please adhere to reporting to CONSORT guidelines and provide a filled-in checklist for your manuscript.

2. Multiple statistical tests were carried out without adjusting for the type 1 error. Please carry out correction methods such as Bonferroni.

3. In the exercise arm, how many physiotherapists were involved in providing advice? 

4. How was the randomization list generated? How was the allocation concealed? Mention these in the methods section.

5. Were the assessors blinded? Mention this in the methods section.

6. Please mention median (ranges) of the VAS scores in the table rather than mean (SD). 

7. Categorization of the pain into different severity scales is more meaningful to the reader and the surgeons rather than numbers and compare the number in each category at baseline and Times T1 and Time 2.

Author Response

Reviewer 4

Comment 1. Please adhere to reporting to CONSORT guidelines and provide a filled-in checklist for your manuscript.

Answer 1: Yes, I did.

Comment 2. Multiple statistical tests were carried out without adjusting for the type 1 error. Please carry out correction methods such as Bonferroni.

Answer 2: We considered the Bonferroni method as a tool for correction of type 1 error, but after a review of critical literature we chose not to use it. In fact, exploration studies such as ours have been recommended not to employ type 1 error adjustment methods in order not to exceed in conservativity. Moreover, using these correction methods has been observed to increase the risk of type 2 error.

For transparency’s sake, I attach the aforementioned references:

Rothman KJ. No adjustments are needed for multiple comparisons. Epidemiology. 1990;1(1):43-46.

Perneger TV. What's wrong with Bonferroni adjustments. BMJ. 1998;316(7139):1236-1238. doi:10.1136/bmj.316.7139.1236.

Armstrong RA. When to use the Bonferroni correction. Ophthalmic Physiol Opt. 2014;34(5):502-508. doi:10.1111/opo.12131.

Comment 3. In the exercise arm, how many physiotherapists were involved in providing advice? 

Answer 3: Only one physiotherapist.

Comment 4. How was the randomization list generated? How was the allocation concealed? Mention these in the methods section.

Answer 4: We addressed this point.

Comment 5. Were the assessors blinded? Mention this in the methods section.

Answer 5: We addressed this point.

Comment 6. Please mention median (ranges) of the VAS scores in the table rather than mean (SD). 

Answer 6: We corrected the reporting of VAS scores as requested.

Comment 7. Categorization of the pain into different severity scales is more meaningful to the reader and the surgeons rather than numbers and compare the number in each category at baseline and Times T1 and Time 2.

Answer 6: Thank you for the input, however for statistical purposes means describe more accurately the behaviour of pain in our sample population. Still, your point is valid, and we addressed it in the paper’s Discussion section.

Round 2

Reviewer 4 Report

Comments and Suggestions for Authors

Thank you for the revision.